# An Innovative Deep Learning Approach for Ventilator-Associated Pneumonia (VAP) Prediction in Intensive Care Units—Pneumonia Risk Evaluation and Diagnostic Intelligence via Computational Technology (PREDICT)

**DOI:** 10.3390/jcm14103380

**Published:** 2025-05-13

**Authors:** Geoffray Agard, Christophe Roman, Christophe Guervilly, Jean-Marie Forel, Véronica Orléans, Damien Barrau, Pascal Auquier, Mustapha Ouladsine, Laurent Boyer, Sami Hraiech

**Affiliations:** 1Service de Médecine Intensive—Réanimation, AP-HM, Hôpital Nord, 13015 Marseille, France; jeanmarie.forel@ap-hm.fr (J.-M.F.); damien.barrau@ap-hm.fr (D.B.); sami.hraiech@ap-hm.fr (S.H.); 2Faculté de Médecine, Aix-Marseille Université, Centre d’Etudes et de Recherches sur les Services de Santé et Qualité de vie (CERESS) EA 3279, 13005 Marseille, France; veronica.orleans@ap-hm.fr (V.O.); pascal.auquier@univ-amu.fr (P.A.); laurent.boyer@ap-hm.fr (L.B.); 3Laboratoire d’Informatique et Système (LIS) UMR 7020 CNRS/AMU/UTLN, Aix Marseille Université, Campus de Saint Jérôme, Bat. Polytech, 13288 Marseille, France; christophe.roman@lis-lab.fr (C.R.); mustapha.ouladsine@lis-lab.fr (M.O.); 4Département d’Information Médicale, AP-HM, Hôpital de la Conception, 13005 Marseille, France

**Keywords:** ventilator-associated pneumonia, artificial intelligence, deep learning, predictive modeling, intensive care, machine learning, MIMIC-IV, time-series analysis, long short-term memory

## Abstract

**Background:** Ventilator-associated pneumonia (VAP) is a common and serious ICU complication, affecting up to 40% of mechanically ventilated patients. The diagnosis of VAP currently relies on retrospective clinical, radiological, and microbiological criteria, which often delays targeted treatment and promotes the overuse of broad-spectrum antibiotics. The early prediction of VAP is crucial to improve outcomes and guide antimicrobial use related to this disease. This study aimed to develop and validate PREDICT (Pneumonia Risk Evaluation and Diagnostic Intelligence via Computational Technology), a deep learning algorithm for early VAP prediction that is based solely on vital signs. **Methods**: We conducted a retrospective cohort study using the MIMIC-IV database, which includes ICU patients who were ventilated for at least 48 h. Five vital signs (respiratory rate, SpO_2_, heart rate, temperature, and mean arterial pressure) were structured into 24 h temporal windows. The PREDICT model, based on a long short-term memory neural network, was trained to predict the onset of VAP 6, 12, and 24 h in the future. Its performance was compared to that of conventional machine learning models (random forest, XGBoost, logistic regression) using their AUPRC, sensitivity, specificity, and predictive values. **Results**: PREDICT achieved high predictive accuracy with AUPRC values of 96.0%, 94.1%, and 94.7% at 6, 12, and 24 h before the onset of VAP, respectively. Its sensitivity and positive predictive values exceeded 85% across all horizons. Traditional ML models showed a drop in performance over longer timeframes. Analysis of the model’s explainability highlighted the respiratory rate, SpO_2_, and temperature as key predictive features. **Conclusions**: PREDICT is the first deep learning model specifically designed for early VAP prediction in ICUs. It represents a promising tool for timely clinical decision-making and improved antibiotic stewardship.

## 1. Introduction

Ventilator-associated pneumonia (VAP) is one of the most common complications in intensive care units (ICUs). From 5% to 40% of patients under invasive mechanical ventilation (MV) are likely to develop at least one VAP during their stay. VAP increases the duration of mechanical ventilation (MV) and the length of ICU stay, leading to a potential increase in mortality of up to 50% [1]. The diagnosis of VAP is based on a combination of three criteria: clinical suspicion, apparition or worsening of radiological infiltrates, and positivity of a respiratory tract culture [2]. Clinical suspicion, the key element of VAP diagnosis, has very low sensitivity and specificity [3,4]. In addition, VAP diagnosis is retrospective by nature, being linked to the positivity of respiratory samples. Current recommendations therefore call for respiratory sampling and probabilistic antibiotic therapy while doctors are awaiting microbiological results to confirm or invalidate the diagnosis [1,2]. Although this approach limits the low sensitivity of markers of clinical suspicion, it exposes the patient to an increased consumption of broad-spectrum antibiotics in the ICU. The early diagnosis of VAP is a critical priority, as delayed or missed diagnoses can lead to prolonged infections and worse patient outcomes, while premature empirical treatments increase the risks of antibiotic resistance and adverse drug effects. The ability to anticipate VAP onset before clinical suspicion arises could enable targeted interventions, reducing unnecessary antibiotic exposure and improving survival rates.

In order to improve the earliness of VAP diagnosis, it is important to compute and merge the information contained in various indicators. The recent advent of artificial intelligence (AI), a set of technologies designed to simulate human cognitive abilities, could make it possible to improve the earliness of VAP diagnosis. To date, the literature on systems for VAP prediction using AI is scarce. Most of these systems use machine learning (ML) algorithms, a branch of AI that uses statistical algorithms to learn from data. Samadani et al. used a ML model for VAP prediction within 24 h, with data on patients’ demographics, vital constants, biology, and mechanical ventilation being used for training [5]. Although the AUROC (75.6%) for their algorithm appeared better than CPIS score, its sensitivity (68%) and specificity (67%) remained unsatisfactory. Meanwhile, deep learning (DL), a specialized branch of machine learning (ML), uses advanced structures called deep neural networks to analyze and interpret data [6,7]. Unlike traditional methods that often require the manual selection of relevant features, DL algorithms can automatically identify and learn important patterns directly from raw data. Additionally, DL is particularly well-suited for processing time-based information, as it can recognize long-term trends and complex sequences in data, making it an ideal tool for tasks that involve changes over time [8]. There is currently no work dealing with the application of deep learning to the prediction of VAPs in the ICU.

The objective of this study was to develop and validate PREDICT (Pneumonia Risk Evaluation and Diagnostic Intelligence via Computational Technology), a deep learning algorithm designed to facilitate the early diagnosis of VAP in ICU patients who have been mechanically ventilated for more than 48 h. Our main hypothesis was that variations in the vital signs of patients, in particular the respiratory rate and SpO_2_, could enable a DL tool to detect VAP occurrence early. Additionally, this study sought to demonstrate that deep learning outperforms traditional machine learning methods in accurately predicting VAP.

## 2. Materials and Methods

### 2.1. Study Design

This was a retrospective cohort study conducted using the publicly available MIMIC-IV (Medical Information Mart for Intensive Care IV) database version 2.2. MIMIC-IV contains de-identified information from ICU admissions at the Beth Israel Deaconess Medical Center between 2008 and 2019 in the United States [9]. This database provides a rich dataset that includes demographic details, vital signs, laboratory results, and treatment information. In this work, to simplify the implementation of a VAP prediction tool in ICUs, we chose to focus solely on vital sign data, which can be directly obtained from monitoring systems. Notably, similar approaches have shown promising results in developing sepsis prediction algorithms using this limited data set [10,11]. The study design followed the Transparent Reporting of a Multivariable Prediction Model for Individual Prognosis or Diagnosis (TRIPOD+AI) guidelines [12] (Appendix A).

The primary objective was to develop and validate a deep learning algorithm for the early prediction of VAP 6, 12 and 24 h in the future using vital sign data.

### 2.2. Patient Population

Patients were included in the study if they were aged 18 years or older, required invasive MV for more than 48 h, and had complete records of vital signs for the study period. Exclusion criteria were defined as the presence of community-acquired pneumonia (as identified by ICD-10 codes or documentation of respiratory infection prior to ICU admission), a MV duration shorter than 48 h, or incomplete or missing key variables that were required for modeling.

The VAP risk period was defined as starting 48 h after the initiation of MV and continuing until 72 h post-extubation. This window was chosen to capture nosocomial pneumonia cases while excluding early-onset pneumonia that was likely acquired before ICU admission.

### 2.3. Data Collection

For the purposes of this study, a MV episode was defined as any continuous period during which a patient received invasive mechanical ventilation, either through an endotracheal tube or a tracheostomy. The start of an MV episode was marked by the initiation of invasive ventilation, and the end was defined as the moment the patient was extubated or transitioned to non-invasive ventilation for more than 48 h [13]. Episodes with interruptions of less than 24 h were considered part of the same continuous ventilation period to account for temporary weaning or procedural pauses commonly seen in ICU settings. Only episodes lasting more than 48 h were included in the analysis. These episodes were extracted from the MIMIC-IV records using specific Structured Query Language (SQL) queries.

Vital signs extracted for this study included respiratory rate, heart rate, mean arterial pressure, body temperature, and oxygen-pulsed saturation (SpO_2_). These variables were also extracted from the MIMIC-IV database’s time-series records of patient monitoring with SQL queries. Additional demographic and clinical variables, such as age, gender, Sepsis-related Organ Failure Assessment (SOFA), and Simplified Acute Physiology Score (SAPS-II) at ICU admission, were extracted from corresponding patient records. ICD-10 codes were used to classify and exclude patients with community-acquired pneumonia and to identify relevant comorbidities, such as chronic obstructive pulmonary disease (COPD), ischemic heart disease, diabetes mellitus, chronic kidney insufficiency, and active cancer. Data on ICU admission sources, ventilation type (invasive or tracheostomy), and duration of MV were obtained from structured database fields. Ventilation-free days (VFDs) D28 refers to the number of days within the first 28 days after the start of MV during which a patient is alive and not dependent on mechanical ventilation.

### 2.4. Outcomes

The primary outcome of this study was the accurate early prediction of VAP within predefined time windows (6, 12, and 24 h) following an observation period of 24 h. Performance was measured using metrics such as the area under the precision–recall curve (AUPRC), sensitivity, positive predictive value (PPV), specificity, and negative predictive value (NPV), which are defined in Appendix A.

Secondary outcomes included a comparison of the models’ performances and assessing the superiority of the deep learning (DL) algorithm (PREDICT) over traditional ML models, including random forest, XGBoost, and logistic regression models, in terms of their predictive accuracy and robustness across all prediction horizons.

### 2.5. Annotation of VAP Events

VAP events were identified based on a standardized methodology using clinical and microbiological criteria, consistent with international guidelines (IDSA/ATS [14] and SFAR/SRLF [15]) and the infection labelling method initially proposed by Seymour et al. [16] and adopted by Samadani et al. [5] in a previous work on VAP prediction (Appendix A).

First, suspected VAP cases were identified by screening MV episodes longer than 48 h and which occurred during the VAP risk period that was previously defined. During this period, respiratory microbiological cultures obtained through bronchial aspirates, bronchoalveolar lavage (BAL), or tracheal aspirates were reviewed for evidence of infection. A positive culture was required to support a diagnosis of pneumonia. Clinical interventions were then assessed in conjunction with microbiological findings. New antibiotic regimens targeting respiratory pathogens were identified if they were initiated within a defined temporal window (72 h before or 24 h after the collection of the microbiological sample).

Additionally, episodes of VAP occurring less than 48 h apart were considered as part of the same infection event and were not treated as independent episodes. The onset of a VAP episode was defined as the earliest timepoint of either the microbiological sample collection or the initiation of new antibiotics.

### 2.6. Selection of Comparator Patients Without VAP

Patients without ventilator-associated pneumonia (VAP) were included as comparators using a 1:1 matching strategy to ensure robust comparisons and minimize confounding. Potential controls were identified among patients with at least one MV episode longer than 48 h that did not meet VAP diagnostic criteria. For each VAP episode, a ventilation episode from a patient without VAP was matched. Matching between these two episodes was stratified based on age, sex, SAPS-II score at ICU admission, and duration of MV. This selection process provided negative examples essential for the machine learning algorithm to distinguish between VAP and non-VAP episodes, improving its specificity.

### 2.7. Data Preprocessing

Preprocessing included several steps to ensure the dataset was suitable for training a predictive model (Appendix A):**(1)** **Data resampling and cleaning**: Vital signs were resampled at hourly scale to standardize time intervals and reduce measurement errors. Missing values were handled using linear interpolation to preserve the continuity and integrity of the time-series data;**(2)** **Normalization**: Each vital sign value was standardized by subtracting the mean and dividing by the standard deviation. This normalization step ensured comparability across variables, preventing any single variable with a larger numerical range (e.g., mean arterial pressure) from disproportionately influencing the algorithm;**(3)** **Temporal windows creation**: To allow the algorithm to analyze time-dependent patterns in the patient data, we divided the continuous flow of vital signs into temporal windows. A temporal window is a defined time segment that contains patient data recorded over a specific period. In this study, each temporal window consisted of 24 h of continuous vital sign recordings.

Each temporal window was composed of two key parts. The observation window represents a 24 h period during which vital signs were collected each hour. The data from this window are the input data used by the algorithm to identify physiological changes. The prediction window is the period following the observation window during which the algorithm predicts the occurrence of VAP. For this study, three prediction windows were defined at 6, 12 and 24 h after the observation window. Temporal windows were created using a sliding window approach. This means that, after constructing a 24 h observation window, we moved it forward by 1 h to create the next window. This overlap ensures the algorithm captures granular temporal patterns without missing key changes in vital signs.

Each temporal window was labeled based on whether a VAP episode occurred within the associated prediction window. If a VAP episode occurred within the prediction window (e.g., at 4 h for a 6 h prediction horizon), this window was labelled as positive. Otherwise, the time window was labelled as negative. This labeling process allowed the algorithm to differentiate between patterns leading to VAP and those not associated with infection.

**(4)** **Balancing the dataset**: Because VAP events were rare in this dataset (less than 1% of temporal windows), the synthetic minority oversampling technique (SMOTE) [17] was applied (Appendix A). This method was used to artificially generate synthetic examples of VAP-positive temporal windows while preserving the structure of the original data. It ensured the model was exposed to sufficient positive examples, enhancing its sensitivity and specificity [18]. Details on the SMOTE technique are available in Appendix A.

### 2.8. Data Splitting for Algorithm Training and Evaluation

The dataset was divided into three subsets: the training set (60%), which was used to train the algorithm; the validation set (20%), which was used to fine-tune hyperparameters and prevent overfitting; and the test set (20%), which was held back for final evaluation of the model’s performance (Appendix A). Stratified sampling was employed to ensure that the proportions of VAP-positive and VAP-negative windows were consistent across all subsets. This step minimized bias and ensured the generalizability of the results.

### 2.9. Algorithm Development

The PREDICT model is based on long short-term memory (LSTM) networks. This type of deep learning architecture is particularly suited for sequential data, as it captures temporal dependencies, such as trends or variations in vital signs over time [19]. The baseline model architecture included 3 LSTM layers, each with 50 cells, followed by a fully connected layer with a sigmoid activation function. A dropout regularization (5–10%) was applied to prevent overfitting. The output of the model was the VAP probability of the prediction window that was considered. Training was conducted using a cross-entropy loss function and the Adam optimizer with adaptive learning rates and employed a stratified cross-validation method with 5 folds (Appendix A). Hyperparameters such as the number of LSTM cells, dropout rate, and learning rate were optimized using the validation set. The best configuration was selected based on the area under the precision–recall curve (AUPRC), a metric particularly well suited for imbalanced datasets [20] (Appendix A).

### 2.10. Comparator Models

To evaluate the added value of deep learning, we trained several traditional machine learning models, including random forest, XGBoost, and logistic regression models. These models were trained and tested using the same dataset and preprocessing pipeline. Their performance was compared to the PREDICT algorithm using metrics such as their sensitivity, specificity, precision, and AUPRC. We compared PREDICT‘s AUPRC with that of other models using a bootstrap resampling approach (*n* = 100 resamples) to determine statistical differences in the AUPRCs.

### 2.11. Model Explainability

To enhance clinical interpretability, we applied the integrated gradients technique [21,22] to analyze the model’s decision-making process. This approach quantified the contribution of each input variable to the prediction, enabling us to identify which vital signs were most influential in detecting VAP.

### 2.12. Model Calibration

To assess the reliability of the predicted probabilities, we produced calibration plots (reliability curves) and computed the Brier score for each prediction horizon (6, 12, and 24 h). The calibration curve compares the average predicted probability to the observed event frequency across bins, reflecting how well predicted risks align with actual outcomes. The Brier score, defined as the mean squared difference between predicted probabilities and true labels, provides a global measure of both calibration and accuracy.

### 2.13. Statistical Analysis

Continuous variables were summarized as medians with interquartile ranges and compared using the Wilcoxon rank-sum test [23], a nonparametric alternative to the *t*-test that is appropriate for non-normally distributed data. Categorical variables were presented as counts and percentages, with differences being assessed using Pearson’s χ^2^ test or Fisher’s exact test. Metrics such as the area under the curve of receiver-operating characteristic (AUROC), precision (predictive positive value), recall (sensitivity), area under the curve of the precision–recall curve (AUPRC), and Youden-index (definitions in Appendix A) were calculated for each model with 95% confidence intervals using bootstrapping (*n* = 100 iterations). The AUPRC was chosen as the primary metric because it accounts for imbalanced datasets better than the AUROC [20]. Deep learning models were constructed using tensorflow (v 2.15) and machine learning models were built with scikit-learn (v 1.5.1) packages. All analyses have been performed in Python language (v 3.11.9), with the pandas (v2.2.2) and numpy (v 1.26.4) packages also being used.

## 3. Results

### 3.1. VAP Episodes

In the MIMIC-IV dataset, between 2008 and 2019, we identified 38,750 invasive MV episodes with 9849 sessions greater than 48 h (25.4%) that concerned 7871 patients (Figure 1). Our VAP annotation algorithm identified 452 VAP episodes for 397 patients and 404 MV episodes > 48 h (4.1%). During their stay in the ICU, 351 patients presented one VAP episode, 41 patients presented two VAP episodes, 1 patient presented three VAP episodes, and 4 patients presented four VAP episodes. The median time from ICU admission to first VAP episode was 6 days (IQR 4–12) and that from MV initiation to the first VAP episode was also 6 days (IQR 3.7–11.3).

### 3.2. Population Characteristics

Despite the stratification, men were significantly more at risk for VAP during their stay (67 vs. 59%—*p =* 0.011). The patients in the VAP group had less ischemic heart disease, chronic kidney insufficiency, and active cancer but more COPD than the no-VAP group. In the study population, 20% of patients were admitted with sepsis, 11% with trauma, and 8.3% with stroke. The main characteristics of the study population are presented in Table 1.

### 3.3. Outcomes—Model Performance

PREDICT was able to predict VAP six hours before onset with a sensitivity of 89.7%, a predictive positive value of 89.8%, and a predictive negative value of 99.7%. The sensitivity and positive predictive values exceeded 85% for all prediction thresholds (Table 2), including 24 h prediction (Sensitivity 85.1%—Specificity 99.2%). Figure 2 shows how the probability of VAP onset at 12 h evolves as a function of the vital signs of a patient who presented several VAPs during his ICU stay. The best structure of the PREDICT algorithm, selected after HPO, was three hidden layers of 50 LSTM cells with a dropout rate of 10% for VAP predictions at 6 and 12 h and 5% for predictions at 24 h (Appendix A). Training curves and a confusion matrix for each prediction threshold are provided in the additional files (Appendix A), as well as all models’ performance metrics that had a confidence interval of 95% (Appendix A).

### 3.4. Comparators Models

Within comparison to the other machine learning algorithms that we trained, PREDICT offers the best AUPRC with, respectively, 96.0%, 94.1%, and 94.7% for VAP prediction at 6, 12, and 24 h. The XGBoost and random forest algorithms provide viable alternatives for early prediction at 6 h, with an AUPRC of 94.1% and 91.7%, respectively. However, their performance rapidly declines for longer-range prediction thresholds until it becomes completely worthless for 24 h predictions. Figure 3 shows the AUPRC and AUROC curves for the PREDICT model compared to the concurrent ML algorithms. The comparison of the AUPRC between the PREDICT and traditional machine learning models revealed a statistically significant difference (*p* < 0.001) for all comparisons.

### 3.5. Calibration Performance

The calibration curves for the three prediction horizons showed good agreement between the predicted probabilities and observed VAP incidence. The Brier scores were 0.04, 0.06, and 0.1, respectively, for the 6 h, 12 h, and 24 h predictions. These results support the good probabilistic calibration of the PREDICT model across all tested time horizons.

### 3.6. PREDICT Model Explainability

The integrated gradients analysis of the PREDICT model reveals how each feature influences predictions over time (Figure 4). On the temporal window scale, the heart rate had a steady, positive contribution, while the mean arterial pressure showed significant early fluctuations. SpO_2_ had a minimal impact, remaining close to zero, whereas the influence of the temperature shifted from negative to a positive toward the end of the sequences. The respiratory rate gained relevance progressively over time. Considering the overall scale, the most critical features for prediction were the SpO_2_, temperature, and respiratory rate, with the mean arterial pressure and heart rate playing smaller roles.

### 3.7. Patients’ Prognosis

The patients in the VAP group had a higher hospital length of stay (25.9 vs. 16.3 days—*p* < 0.001) and ICU length of stay (18.9 vs. 8.4 days—*p* < 0.001), as well as an increased mortality in ICU. The duration of MV was increased in the VAP group and the number of ventilation-free days at day 28 was also significantly lower. Outcomes for the population are described in Table 3.

## 4. Discussion

In this study, we developed and validated PREDICT, a new innovative deep learning model for the early detection of VAP 6, 12, and 24 h in the future. PREDICT demonstrated excellent diagnostic performance, with an AUROC of 99% for all prediction thresholds. Its sensitivity (Se) and positive predictive value (PPV) ranged from 85.1% to 89.7%, while the AUPRC exceeded 94%, indicating a high predictive capability for both the presence and absence of VAP.

PREDICT represents the first application of deep learning to VAP prediction in ICUs. Previous studies using machine learning algorithms achieved a lower performance than PREDICT. Liang et al. developed a random forest model to predict VAP within 24 h of intubation using around 30 demographic, biological, and physiological variables [24].

While this model showed satisfactory performance (AUROC 84%, Se 74%, specificity 71%), its reliance on diverse and inconsistent data sources limited its ICU applicability. Similarly, Samadani et al. used an XGBoost model, incorporating a wide range of variables to predict VAP within 24 h [5]. Although encouraging (AUROC 76%, AUPRC 75%), this model had low sensitivity (67.5%) and PPV (68.5%) scores, restricting its clinical utility. In contrast, PREDICT outperformed these models across all prediction horizons and maintained a stable predictive performance even for longer intervals, highlighting the advantage of deep learning in modeling complex relationships without extensive manual feature engineering [7].

To ensure comprehensive evaluation, we compared PREDICT with random forest and XGBoost models. While these traditional algorithms closely matched PREDICT’s performance for short prediction tasks (6 h), with AUROCs of 99% and AUPRCs between 92% and 94%, their accuracy declined for longer prediction horizons, unlike PREDICT, which remained robust. The stability of PREDICT’s predictions illustrates the strength of deep learning in extracting subtle patterns from time-series data, which allows for more reliable and generalizable predictions. PREDICT outperformed conventional ML models, likely due to its capacity to capture nonlinear temporal dependencies within sequential vital signs. Its LSTM-based architecture allows dynamic feature learning across time, in contrast to the static nature of tree-based models.

PREDICT offers physicians a valuable tool for the early identification of VAP, allowing intervention from 6 to 24 h before clinical suspicion arises. This early detection can enable the timely initiation of targeted antibiotics, potentially reducing the progression of issues to severe infections, organ dysfunction, and prolonged ICU stays [25]. Gaining 6–24 h is crucial in critically ill patients, where early intervention can prevent progression to severe infections, organ dysfunction, and prolonged ICU stays [26]. PREDICT’s high negative predictive value can also help in optimizing antimicrobial stewardship by avoiding the unnecessary use of broad-spectrum antibiotics, thus reducing antibiotic resistance risks [27]. In such cases, it can prompt the physician to investigate for another origin of sepsis than the lungs. Future work will investigate intermediate prediction windows, such as 18 h windows, which may offer an optimal trade-off between early detection and clinical actionability. This additional time horizon could help refine the temporal resolution of the model and further support its clinical integration. The fact that PREDICT only uses vital constants that are available on ICU monitors gives us hope that, in the future, such technology can be integrated directly into these monitors, making it much easier for clinicians to use. Indeed, the model’s compact architecture achieves inference times on the order of a few seconds, making bedside application feasible with modest computational resources. However, although our algorithm offers promising predictive capabilities, its use must be carefully integrated into clinical workflows to avoid promoting unnecessary antibiotic use. Model outputs should support, not replace, clinical judgment, and guidelines for interpreting alerts will be necessary to prevent overprescription and resistance development. Rather than directly prompting antibiotic initiation, a high-risk prediction should lead to clinical reassessment and the consideration of additional diagnostics, such as respiratory sampling. Stewardship protocols must frame its use to avoid overtreatment or unnecessary antimicrobial exposure.

A key barrier to the adoption of deep learning in medicine has been the lack of explainability. Trust in AI systems depends on understanding the reasoning behind their predictions, especially in the medical field [28,29]. Recent advancements, such as the integrated gradient technique [21], have improved their interpretability. By applying this method, we highlighted the contribution of individual variables to the model’s predictions, identifying changes in temperature and the respiratory rate as key indicators of VAP. These findings align with earlier studies which show the predictive value of these variables in machine learning models [5,24].

It may seem paradoxical that PREDICT, despite using fewer variables, outperforms models that incorporate dozens of features. Complex models often suffer from overfitting [30], capturing noise along with trends, which hampers their generalization to new data. Moreover, multicollinearity between variables can dilute the predictive signal, further reducing the model’s performance [31]. Similar observations were made by Desautels et al., who achieved promising sepsis prediction results (Se 80%, Sp 80%, AUROC 88%) using only the vital signs, SpO_2_, Glasgow score, and age of patients, demonstrating that simpler models can sometimes yield superior results [10]. In this work, we deliberately restricted the model to five basic vital signs to enhance its feasibility for real-time deployment that does not rely on comprehensive data integration. While this design choice favors simplicity and broad applicability, it may limit the model’s ability to capture additional clinical signals that are available from laboratory, imaging, or ventilatory data, which could be explored in future model iterations.

Our work also stands out due to its annotation technique, adapted from Seymour et al. and previously used by Samadani et al., which adheres to international recommendations [5,14,15,16]. This method ensures microbiological accuracy and resolves key challenges in annotating VAP, such as variability in definitions. ICD codes, often used for annotation, lack both sensitivity (59%) and specificity (PPV 27–42%), making them unsuitable for precise classification [32]. By employing time windows to pinpoint the diagnosis of VAP, we avoided the limitations of retrospective ICD coding, which fails to capture the exact timing of clinical events.

A primary limitation of this study lies in its monocentric design, relying exclusively on the MIMIC-IV database, which may limit the generalizability of PREDICT to diverse ICU settings. The algorithm learning was initially based on a single-center dataset as a proof of concept to assess the predictive value of vital signs alone using deep learning. Additionally, the data that are available in MIMIC-IV may not fully capture the complexity of clinical decisions. To overcome these challenges, future work should include retraining the model with data from multiple hospitals, incorporating diverse patient populations and clinical practices. Another limitation of our study is that we did not adjust for potential variations in VAP-prevention practices over the 2008–2019 period. Changes in clinical protocols, such as the increased use of ventilator care bundles (including head-of-bed elevation, sedation interruption, oral care, and prophylaxis strategies), could theoretically influence the incidence of VAP and thus affect the model’s learning. However, a recent analysis of the MIMIC-IV cohort by Leong et al. (2024) [33] showed that compliance with VAP prevention bundles was consistently low across this period, with only head-of-bed elevation being routinely applied. Compliance to the full bundle remained extremely rare (<1%), suggesting that systematic practice changes were limited during this study’s timeframe. Nonetheless, we recognize that even small shifts in clinical behavior could introduce residual confounding, and this should be taken into account when interpreting the model’s generalizability. The question also arises of assessing the impact on the model of changes in resuscitation practices during this period. We did not adjust for temporal trends, which could introduce bias if clinical changes over time influenced the incidence of VAP or diagnostic patterns. However, the exclusive use of physiological signals may mitigate this effect, as these core variables remain consistent across practice eras. Our model also did not adjust for prior antibiotic use before the onset of VAP. Although antibiotic exposure could delay the microbiological confirmation of infection, PREDICT is based solely on vital sign patterns and aims to detect the physiological manifestations of VAP. Nonetheless, prior antibiotic therapy could introduce some diagnostic uncertainty, which may impact the model’s annotation accuracy and which represents a potential source of bias. Although both endotracheal and tracheostomized patients were included in the training cohort, we did not account for the airway type or ventilation mode in the model. While these factors are known to influence VAP risk, our objective was to evaluate the predictive value of physiological signals alone. Future work will explore whether incorporating these variables can further improve the model’s prediction accuracy. It is also important to note that, due to the use of rolling time windows, some observation periods were located close to the annotated VAP event, possibly capturing early manifestations of infection. While this could lead to a detection effect, the model was also trained on earlier windows that were distributed across a wide temporal range. This variability helps reduce bias, but further sensitivity analyses are warranted to assess the model’s performance across different prediction distances.

Finally, while PREDICT has shown potential to improve early VAP detection, its clinical benefits must be confirmed through prospective randomized controlled trials to assess its impact on clinical outcomes, such as mortality, the duration of ventilation, and antibiotic exposure. A high predictive accuracy alone does not ensure clinical benefit, and there is a risk that the systematic application of predictive alerts could inadvertently lead to overtreatment or prolong instances of mechanical ventilation if it is not integrated with appropriate stewardship strategies. Therefore, beyond demonstrating technical performance, future prospective studies must evaluate whether PREDICT effectively reduces ventilation days, optimizes antibiotic use, and improves patient-centered outcomes without introducing new harms.

## 5. Conclusions

In this work, we developed and internally validated a new deep-learning model for VAP prediction using the MIMIC-IV cohort. Using a small number of clinical variables, our PREDICT algorithm demonstrated a high capacity for detecting both short- and long-term VAP. The superiority of the deep-learning approach was confirmed by comparison with other machine learning algorithms trained in the same way. An external validation of the algorithm and its implementation in a prospective trial strategy should be considered in future work.

## Figures and Tables

**Figure 1 jcm-14-03380-f001:**
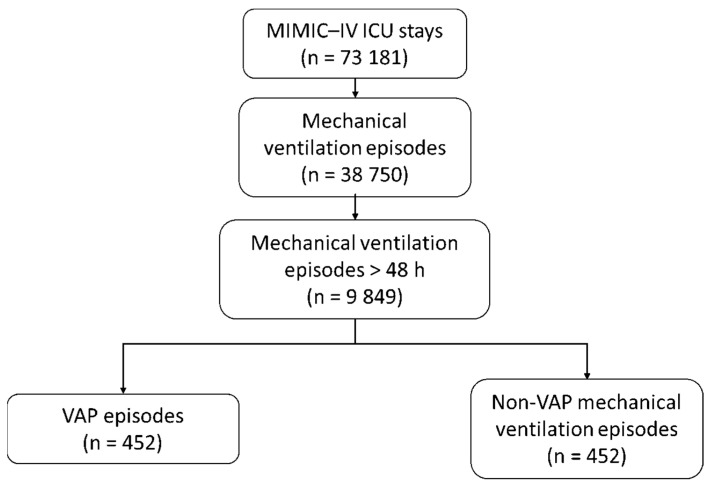
Flow chart.

**Figure 2 jcm-14-03380-f002:**
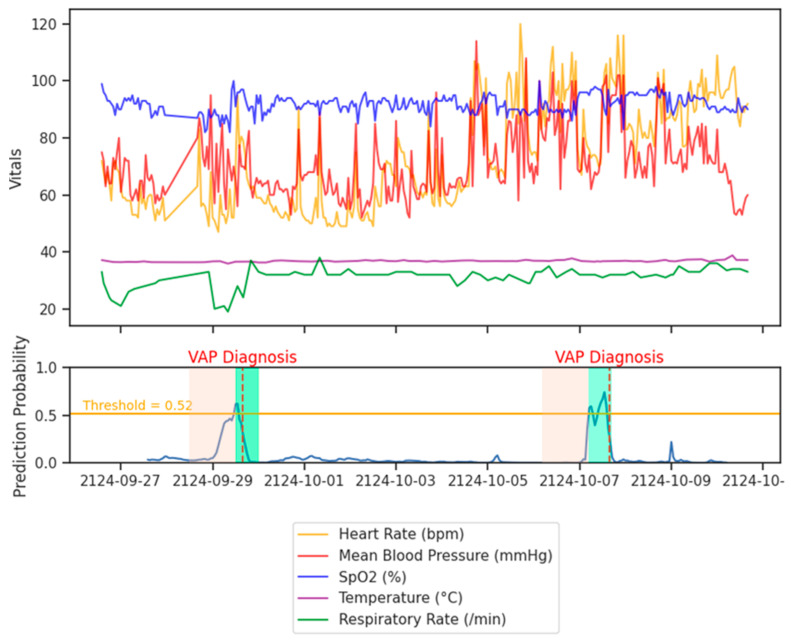
Patient vital signs (**top**) and the predicted probability of VAP 12 h in the future (**bottom**) over time for a patient with VAP during their ICU stay. The orange shaded area represents the 24 h observation window, the green shaded area marks the 12 h prediction window. Red dashed lines indicate the clinical VAP diagnosis times. The algorithm predicts a future VAP occurrence when the probability crosses the threshold (orange line).

**Figure 3 jcm-14-03380-f003:**
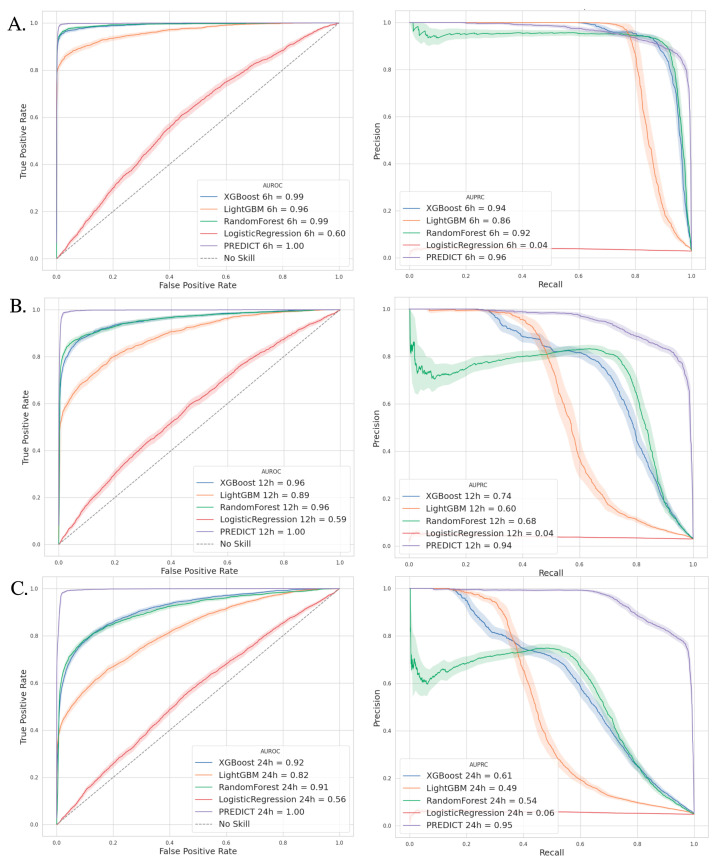
ROC and precision–recall curves for all algorithms. (**A**) On left, receiver operating characteristic (ROC) curves with corresponding area under the curve (AUROC) for all algorithms for VAP prediction 6 h in the future. On right, precision–recall curves with corresponding area under the curve (AUPRC) for all algorithms with VAP prediction 6 h in the future. (**B**) VAP prediction 12 h in the future. (**C**) VAP prediction 24 h in the future. Please note that the reported AUROC values were rounded for clarity. Exact values (e.g., 0.998) are slightly below 1 and are shown with greater precision in the Appendix A.

**Figure 4 jcm-14-03380-f004:**
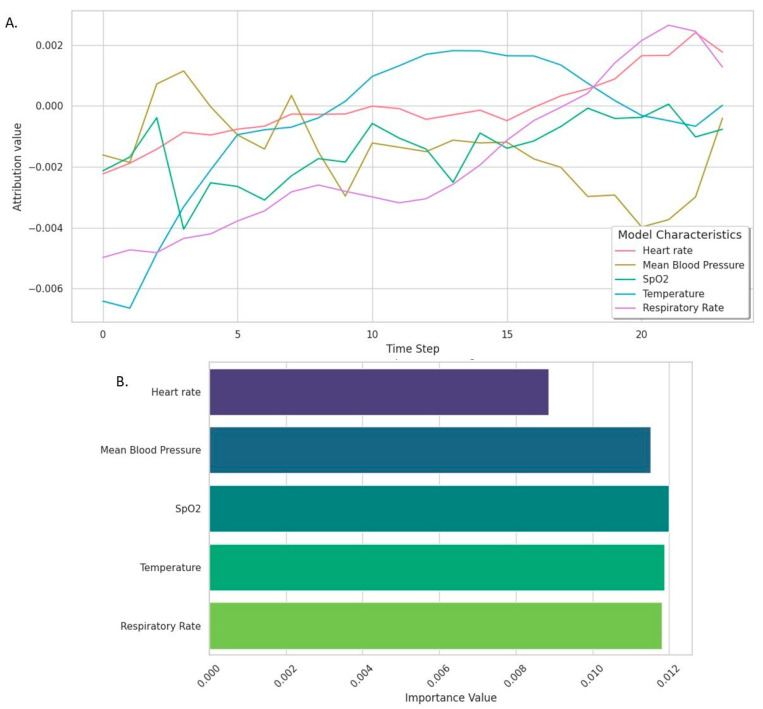
PREDICT model explainability. (**A**) Integrated gradient attributions in temporal window scale. Integrated attributions measure the importance of input features for model prediction. High attribution values indicate that the feature had a significant impact on the prediction of the target class. (**B**) Feature importance in model overall decision.

**Table 1 jcm-14-03380-t001:** Patients’ characteristics.

	Overall ^1^, *n* = 904	VAP ^1^, *n* = 452	No VAP ^1^, *n* = 452	*p*-Value ^2^
**Sex (Male)**	573 (63%)	305 (67%)	268 (59%)	**0.011 ***
**Age (years)**	64.2 [52.1–75.3]	63.9 [50.7–74.3]	65.1 [52.9–76.1]	0.2
**Pre-existing Diseases *n* (%)**				
Hypertension	450 (50%)	217 (48%)	233 (52%)	0.3
Ischemic heart disease	306 (34%)	131 (29%)	175 (39%)	**0.002 ***
Diabetes mellitus	216 (24%)	106 (23%)	110 (24%)	0.8
Chronic renal failure	218 (24%)	95 (21%)	123 (27%)	**0.029 ***
Obstructive sleep apnea	145 (16%)	69 (15%)	76 (17%)	0.5
Active cancer	134 (15%)	56 (12%)	78 (17%)	**0.039 ***
COPD	66 (7.3%)	43 (9.5%)	23 (5.1%)	**0.011 ***
Active hematological malignancy	31 (3.4%)	18 (4.0%)	13 (2.9%)	0.4
**Source of admission to ICU**				0.9
Emergency ward	665 (74%)	331 (73%)	334 (74%)	
Medical ward	150 (17%)	74 (16%)	76 (17%)	
Elective surgery	89 (9.8%)	47 (10%)	42 (9.3%)	
**SOFA—admission**	2.0 [0.0–4.0]	1.0 [0.0–4.0]	2.0 [0.0–4.0]	0.2
**SAPS-II on admission**	42.0 [32.0–54.0]	42.0 [31.0–53.0]	43.0 [34.0–55.0]	0.053
**Time from ICU admission to initiation of MV(hours)**	7.2 [1.7–67.3]	8.2 [1.8–73.0]	6.0 [1.7–60.4]	0.4
**Reason for ICU admission**				
Sepsis	179 (20%)	91 (20%)	88 (19%)	
Trauma	97 (11%)	73 (16%)	24 (5.3%)	
Hemorrhagic or ischemic stroke	75 (8.3%)	44 (9.7%)	31 (6.9%)	
Acute malignancy	46 (5.1%)	17 (3.8%)	29 (6.4%)	
ARDS	40 (4.4%)	34 (7.5%)	6 (1.3%)	
Pneumonia	34 (3.8%)	20 (4.4%)	14 (3.1%)	
Myocardial infarction	26 (2.9%)	8 (1.8%)	18 (4.0%)	

^1^ *n* (%); median [25–75%] ^2^ Pearson’s chi-squared test; Wilcoxon rank sum; COPD: chronic obstructive pulmonary disease; ICU: intensive care unit; MV: mechanical ventilation; ARDS: acute respiratory distress syndrome. * *p*-value < 0.05.

**Table 2 jcm-14-03380-t002:** PREDICT model performance for different VAP prediction thresholds. Best classification threshold was calculated for sensibility = PPV (precision = recall). Confidence intervals of 95% are available in the additional files.

	VAP Prediction	Best Threshold	AUPRC (%)	Sensibility (%)	Specificity (%)	PPV (%)	NPV (%)
**PREDICT Model**	6 h	0.53	96.0	89.7	99.7	89.8	99.7
	12 h	0.52	94.1	85.9	99.6	85.6	99.6
	24 h	0.43	94.7	85.1	99.2	85.0	99.2

AUPRC: area under the precision–recall curve; PPV: predictive positive value; NPV: predictive negative value.

**Table 3 jcm-14-03380-t003:** Patient prognosis outcomes.

	Overall ^1^, *n* = 904	VAP ^1^, *n* = 452	No VAP ^1^, *n* = 452	*p*-Value ^2^
**Length of stay—Hospital** (days)	21.4 [12.9–35.2]	25.9 [17.1–39.0]	16.3 [9.7–28.3]	**<0.001 ***
**Length of stay—ICU** (days)	14.0 [7.8–22.9]	18.9 [13.0–30.0]	8.4 [5.3–15.7]	**<0.001 ***
**Time from ICU admission to death** (days)	38.0 [13.5–117.5]	42.5 [18.3–103.3]	30.1 [7.7–147.8]	**0.007 ***
**Duration of mechanical ventilation** (days)	8.4 [4.0–15.4]	13.6 [8.4–21.7]	4.4 [3.0–8.3]	**<0.001 ***
**Ventilation-free days D28** (days)	14.5 [0.0–22.6]	9.1 [0.0–17.6]	20.9 [0.0–24.5]	**<0.001 ***
**ICU mortality**	188 (21%)	110 (24%)	78 (17%)	**0.009 ***
**In-hospital mortality**	261 (29%)	139 (31%)	122 (27%)	0.2

^1^ *n* (%); median [25–75%]; ^2^ Pearson’s chi-squared test; Wilcoxon rank sum test. * *p*-value < 0.05.

## Data Availability

The dataset used in this study is publicly available through the MIMIC-IV database, hosted by the Massachusetts Institute of Technology (MIT) at https://physionet.org/content/mimiciv/ (accessed on 12 October 2023). Access to MIMIC-IV requires completion of the appropriate data use agreement and certification in research ethics training. Code for this project is available on request from the first author (G.A.).

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
