# Peer review of "An Innovative Deep Learning Approach for Ventilator-Associated Pneumonia (VAP) Prediction in Intensive Care Units—Pneumonia Risk Evaluation and Diagnostic Intelligence via Computational Technology (PREDICT)"

_jcm, 2025, doi:10.3390/jcm14103380_

Round 1

Reviewer 1 Report

Comments and Suggestions for Authors
  1. Did the authors consider different VAP prevention protocols (such as head-of-bed elevation, oral care, or sedation protocols) used in ICU during the years 2008 to 2019? These can reduce VAP, and if they changed over the 11 years of data, the prediction might not be accurate. The authors did not adjust for this bias in clinical practice variation. They must explain this or add it as a limitation.
  2. Was there any adjustment or stratification based on prior antibiotic use before VAP onset? If patients have already got antibiotics, it could delay or mask VAP, changing the model's ability to predict real infections. But the study did not adjust for this, which brings antibiotic exposure bias. Authors need to include this point or explain why this data was not used.
  3. The study does not say how the model’s predictions will be used. A tool that gives frequent VAP alerts could make doctors give more antibiotics, not less. This could create over-prescription problems and resistance. The authors should suggest a guideline on how to respond to the model’s alerts or state this as a risk.
  4. Why did the authors only use five basic vital signs? Deep learning is powerful and can learn from more data, like labs or X-rays. By using only vitals they risk underfitting and missing important clinical signals. They should explain this choice more clearly or call it a model limitation.
  5. Can the PREDICT model run on an ICU monitor or on a hospital server? There is no mention of model size, computing time or memory use. These are technical barriers that the authors must describe or admit as future work needed.
  6. How the model improves key ICU goals, like reducing ventilation days or saving antibiotics? The study shows high prediction accuracy but no real outcome changes. So, it’s hard to the readers to know if the model helps in real life. This gap must be noted as a limitation and need for future prospective testing.
  7. Did the model account for type of airway access (endotracheal vs tracheostomy) and ventilation mode? Tracheostomy versus endotracheal tube and different ventilator settings affect lung infections. But the study doesn’t include this. It’s a miss because ventilation type bias is well-known in VAP studies. The authors must add this missing factor to the limitations.
  8. How did the authors handle temporal trends in ICU care over 11 years of data, from 2008 to 2019? Medical care changed a lot during this time, and this could affect VAP rates. If this was not accounted for, the model might be learning patterns from old practice not current ones. The authors must say whether they tested for secular trends. If not they must admit this as source of bias.
  9. Why did the authors not perform an external validation on another dataset? The model was trained and tested only on MIMIC-IV, from one hospital. We don’t know if it works in Europe, Asia, or even another American hospital. This is a major flaw. The authors must say this is a future step and add it clearly in the discussion as a limitation.
  10. The authors did not present calibration plots or Brier scores. They used only AUPRC and AUROC. These metrics show if the model can rank patients correctly, but not if the predicted probability is true. For example, a model can say 80% VAP risk when it’s only 20%. Without calibration, this is misleading. Calibration is especially important in clinical models. The authors must run calibration plots (reliability curves e.g.) and Brier scores. If they cannot do this now they must say it clearly in the statistical limitations and plan for it in future versions.
  11. How did the authors control for label leakage between the observation and prediction windows? If signs of early VAP already appear in the observation window, and the model predicts VAP in the next 6 hours, then it’s not really “predicting” the future, it’s just detecting the present. This causes over-optimistic results. The authors must clarify how they avoided this maybe with sensitivity analysis. If not done this is a methodological flaw and must be written as a study limitation.

Author Response

1. Did the authors consider different VAP prevention protocols (such as head-of-bed elevation, oral care, or sedation protocols) used in ICU during the years 2008 to 2019? These can reduce VAP, and if they changed over the 11 years of data, the prediction might not be accurate. The authors did not adjust for this bias in clinical practice variation. They must explain this or add it as a limitation.

We thank the reviewer for highlighting this important point. Compliance with ventilator-associated pneumonia (VAP) prevention bundles in the ICU indeed represents a potential source of bias over the long study period (2008–2019).

To address this concern, we reviewed recent literature analyzing the MIMIC-IV cohort, notably the study by Leong et al. (2024) [Leong YH, et al., Anesthesiology and Perioperative Science], which specifically evaluated compliance to VAP prevention protocols within the MIMIC-IV database. Their results demonstrated that compliance to the Institute for Healthcare Improvement (IHI) ventilator care bundle was extremely low overall — with only 0.3% of patients achieving full bundle compliance, and only head-of-bed elevation being routinely implemented (89% compliance), while other interventions such as sedation protocols, oral care, or prophylaxis measures had very low adherence rates.

These findings suggest that throughout the 2008–2019 period, major changes or systematic improvements in VAP prevention bundle compliance were unlikely to significantly impact the general incidence of VAP recorded in MIMIC-IV. Consequently, the low and stable compliance likely limits the bias introduced by evolving prevention protocols on our predictive model.

However, we acknowledge that even low variations in clinical practice could influence VAP risk to some extent. Therefore, we have added a discussion of this point in the limitations section of the revised manuscript, explicitly stating that variations in bundle compliance and ICU protocols over time represent a potential residual confounder that could not be fully adjusted for in our model.

Reference added:
Leong YH, Khoo YL, Abdullah HR, Ke Y. Compliance to ventilator care bundles and its association with ventilator-associated pneumonia. Anesthesiology and Perioperative Science. 2024;2:20. doi:10.1007/s44254-024-00059-1.

Addition to the manuscript in the discussion section :

“Another limitation of our study is that we did not adjust for potential variations in VAP prevention practices over the 2008–2019 period. Changes in clinical protocols, such as increased use of ventilator care bundles (including head-of-bed elevation, sedation interruption, oral care, and prophylaxis strategies), could theoretically influence VAP incidence and thus affect the model’s learning. However, a recent analysis of the MIMIC-IV cohort by Leong et al. (2024) [31] showed that compliance with VAP prevention bundles was consistently low across this period, with only head-of-bed elevation being routinely applied. Compliance to the full bundle remained extremely rare (<1%), suggesting that systematic practice changes were limited during the study timeframe. Nonetheless, we recognize that even small shifts in clinical behavior could introduce residual confounding, and this should be taken into account when interpreting the model's generalizability.”

2. Was there any adjustment or stratification based on prior antibiotic use before VAP onset? If patients have already got antibiotics, it could delay or mask VAP, changing the model's ability to predict real infections. But the study did not adjust for this, which brings antibiotic exposure bias. Authors need to include this point or explain why this data was not used.

We thank the reviewer for this insightful comment. We acknowledge that prior antibiotic exposure could potentially influence the timing of microbiological confirmation of VAP. However, our model is specifically based on the analysis of variations in vital signs — such as respiratory rate, oxygen saturation, temperature, heart rate, and mean arterial pressure — rather than on microbiological or therapeutic data.

The goal of the PREDICT model is to detect clinical patterns suggestive of the physiological onset of pneumonia, independently of microbiological documentation or antibiotic administration. Even if antibiotic exposure delays bacterial growth or masks microbiological confirmation, the underlying physiological alterations typically associated with VAP (such as increased respiratory rate, impaired oxygenation, and temperature variations) would still develop and be detectable through vital sign monitoring.

Therefore, while prior antibiotic use may affect the timing of traditional VAP diagnosis, it is unlikely to alter the emergence of the clinical patterns that the model is trained to recognize. For these reasons, we did not adjust for prior antibiotic exposure in the model. Nonetheless, we have added a sentence in the limitations section to acknowledge that antibiotic exposure could introduce a degree of diagnostic uncertainty, which may impact annotation but not necessarily the clinical signal detection.

Addition to the manuscript in the discussion section :

“Our model did not adjust for prior antibiotic use before VAP onset. Although antibiotic exposure could delay microbiological confirmation of infection, PREDICT is based solely on vital sign patterns and aims to detect the physiological manifestations of VAP. Nonetheless, prior antibiotic therapy could introduce some diagnostic uncertainty, which may impact annotation accuracy and represents a potential source of bias.”

3. The study does not say how the model’s predictions will be used. A tool that gives frequent VAP alerts could make doctors give more antibiotics, not less. This could create over-prescription problems and resistance. The authors should suggest a guideline on how to respond to the model’s alerts or state this as a risk.

We appreciate this important point. We agree that inappropriate or excessive use of the model's alerts could paradoxically increase antibiotic prescription. As suggested, we added in the discussion that any clinical deployment of the model should be accompanied by strict diagnostic stewardship protocols to mitigate this risk, and that uncontrolled use could worsen antimicrobial resistance.

Addition to the manuscript in the discussion section :

“But although our algorithm offers promising predictive capabilities, its use must be carefully integrated into clinical workflows to avoid promoting unnecessary antibiotic use. Model outputs should support, not replace, clinical judgment, and guidelines for interpreting alerts will be necessary to prevent overprescription and resistance development. Rather than directly prompting antibiotic initiation, a high-risk prediction should lead to clinical reassessment and consideration of additional diagnostics, such as respiratory sampling. Stewardship protocols must frame its use to avoid overtreatment or unnecessary antimicrobial exposure.

4. Why did the authors only use five basic vital signs? Deep learning is powerful and can learn from more data, like labs or X-rays. By using only vitals they risk underfitting and missing important clinical signals. They should explain this choice more clearly or call it a model limitation.

Our choice to use only five basic vital signs was intentional. The objective was to develop a simple, easily implementable model based solely on routinely monitored parameters, without needing additional laboratory or imaging data. This design choice aimed to facilitate bedside integration. However, we acknowledge that the exclusion of broader clinical data could limit the model’s predictive potential, and we add some insight about it in the discussion section.

Addition to the manuscript in the discussion section :

“In this work we deliberately restricted the model to five basic vital signs to enhance feasibility for real-time deployment without relying on comprehensive data integration. While this design choice favors simplicity and broad applicability, it may limit the model’s ability to capture additional clinical signals available from laboratory, imaging, or ventilatory data, which could be explored in future model iterations.”

5. Can the PREDICT model run on an ICU monitor or on a hospital server? There is no mention of model size, computing time or memory use. These are technical barriers that the authors must describe or admit as future work needed.

We thank the reviewer for this comment. The PREDICT model is relatively lightweight, and inference times are compatible with real-time bedside application. However, we agree that a formal evaluation of computational requirements and latency was not performed in this study. We added a sentence in the discussion section to assess the feasibility of implementing the algorithm at the patient's bedside

Addition to the manuscript in the discussion section :

“Indeed, the model’s compact architecture achieves inference times on the order of a few seconds, making bedside application feasible with modest computational resources.”

6. How the model improves key ICU goals, like reducing ventilation days or saving antibiotics? The study shows high prediction accuracy but no real outcome changes. So, it’s hard to the readers to know if the model helps in real life. This gap must be noted as a limitation and need for future prospective testing.

We recognize that although PREDICT shows excellent predictive performance, its actual impact on clinical outcomes (ventilation duration, antibiotic consumption, mortality) remains untested. We have added in the discussion that future prospective interventional studies are necessary to demonstrate real-world clinical benefits.

Addition to the manuscript in the discussion section :

“High predictive accuracy alone does not ensure clinical benefit, and there is a risk that systematic application of predictive alerts could inadvertently lead to overtreatment or prolong mechanical ventilation if not integrated with appropriate stewardship strategies. Therefore, beyond demonstrating technical performance, future prospective studies must evaluate whether PREDICT effectively reduces ventilation days, optimizes antibiotic use, and improves patient-centered outcomes without introducing new harms.”

7. Did the model account for type of airway access (endotracheal vs tracheostomy) and ventilation mode? Tracheostomy versus endotracheal tube and different ventilator settings affect lung infections. But the study doesn’t include this. It’s a miss because ventilation type bias is well-known in VAP studies. The authors must add this missing factor to the limitations.

We thank the reviewer for this important observation. The training cohort included patients who underwent both forms of invasive ventilation — via endotracheal tube and tracheostomy — reflecting real-world ICU populations. While we agree that both airway interface and ventilation mode may influence the risk and pathophysiology of VAP, these variables were not incorporated into the model, primarily to maintain the focus on physiological signal analysis and maximize implementation simplicity. The central aim of this study was to assess whether variations in basic vital signs alone could predict VAP with high accuracy, which was confirmed by the model's strong performance metrics. Nevertheless, we recognize the potential added value of these clinical parameters, and we plan to explore their contribution in future work by integrating airway interface and ventilation mode into subsequent model iterations. A corresponding statement has been added to the Limitations section of the manuscript.

Addition to the manuscript in the discussion section :

“Although both endotracheal and tracheostomized patients were included in the training cohort, we did not account for airway type or ventilation mode in the model. While these factors are known to influence VAP risk, our objective was to evaluate the predictive value of physiological signals alone. Future work will explore whether incorporating these variables can further improve prediction accuracy.”

8. How did the authors handle temporal trends in ICU care over 11 years of data, from 2008 to 2019? Medical care changed a lot during this time, and this could affect VAP rates. If this was not accounted for, the model might be learning patterns from old practice not current ones. The authors must say whether they tested for secular trends. If not they must admit this as source of bias.

We thank the reviewer for this relevant observation. Our dataset spans an 11-year period (2008–2019), during which clinical practices in ICU may have evolved, including VAP prevention strategies, antibiotic stewardship, and ventilator management. We did not include temporal variables or stratify model training by admission year, and we acknowledge that secular trends could introduce bias, especially if patterns learned by the model reflect outdated practices. However, the choice to rely exclusively on vital signs — core physiological data — was intended to minimize dependence on temporally variable practices. We have now added this point explicitly as a limitation in the revised manuscript.

Addition to the manuscript in the discussion section :

“The question also arises of assessing the impact on the model of changes in resuscitation practices during this period. We did not adjust for temporal trends, which could introduce bias if clinical changes over time influenced VAP incidence or diagnostic patterns. However, the exclusive use of physiological signals may mitigate this effect, as these core variables remain consistent across practice eras.”

9. Why did the authors not perform an external validation on another dataset? The model was trained and tested only on MIMIC-IV, from one hospital. We don’t know if it works in Europe, Asia, or even another American hospital. This is a major flaw. The authors must say this is a future step and add it clearly in the discussion as a limitation.

We thank the reviewer for this important point. The initial objective of this study was to establish a proof of concept demonstrating that temporal variations in vital signs alone could serve as a reliable basis for early VAP prediction using deep learning. To that end, we focused on a single-center dataset (MIMIC-IV) to ensure data consistency and reduce confounding variability in this initial phase. However, we fully acknowledge that the monocentric nature of the dataset limits the generalizability of our findings. External and multicenter validation will be essential to assess the robustness and transferability of the model across diverse patient populations and care settings. This necessity has now been clearly stated in the revised manuscript.

Addition to the manuscript in the discussion section :

Algorithm learning was initially based on a single-center dataset as a proof of concept to assess the predictive value of vital signs alone using deep learning. Additionally, the data available in MIMIC-IV may not fully capture the complexity of clinical decisions. To overcome these challenges, future work should include retraining the model with data from multiple hospitals, incorporating diverse patient populations and clinical practices.

10. The authors did not present calibration plots or Brier scores. They used only AUPRC and AUROC. These metrics show if the model can rank patients correctly, but not if the predicted probability is true. For example, a model can say 80% VAP risk when it’s only 20%. Without calibration, this is misleading. Calibration is especially important in clinical models. The authors must run calibration plots (reliability curves e.g.) and Brier scores. If they cannot do this now they must say it clearly in the statistical limitations and plan for it in future versions.

We thank the reviewer for this important remark. In response, we have computed and included calibration plots (reliability curves) and Brier scores for the three prediction horizons (6, 12, and 24 hours). These analyses provide additional insight into the reliability of the predicted probabilities and complement the discrimination metrics already reported. The corresponding methodology and results have been added to the manuscript.

Addition to the manuscript in the material and methods section :

“Model Calibration

To assess the reliability of the predicted probabilities, we produced calibration plots (reliability curves) and computed the Brier score for each prediction horizon (6, 12, and 24 hours). The calibration curve compares the average predicted probability to the observed event frequency across bins, reflecting how well predicted risks align with actual out-comes. The Brier score, defined as the mean squared difference between predicted prob-abilities and true labels, provides a global measure of both calibration and accuracy.”

Addition to the manuscript in the result section :

“Calibration performance

Calibration curves for the three prediction horizons showed good agreement be-tween predicted probabilities and observed VAP incidence. The Brier scores were 0.04, 0.06 and 0.1 respectively for the 6h, 12h and 24 hour prediction. These results support the good probabilistic calibration of the PREDICT model across all tested time horizons.”

Addition to the manuscript in the appendix D section :

Figure D7. Calibration plot for PREDICT algorithm for 6h VAP prediction

Figure D8. Calibration plot for PREDICT algorithm for 12h VAP prediction

Figure D9. Calibration plot for PREDICT algorithm for 24h VAP prediction

11. How did the authors control for label leakage between the observation and prediction windows? If signs of early VAP already appear in the observation window, and the model predicts VAP in the next 6 hours, then it’s not really “predicting” the future, it’s just detecting the present. This causes over-optimistic results. The authors must clarify how they avoided this maybe with sensitivity analysis. If not done this is a methodological flaw and must be written as a study limitation.

We thank the reviewer for raising this important point. Due to the use of overlapping sliding temporal windows, some observation windows inevitably occur very close to the annotated VAP onset and may include early physiological changes associated with infection. However, our training set also includes a wide range of windows located much earlier in time often several hours before the VAP event. This distribution allows the model to learn patterns from both early and late phases of infection progression, reducing the risk that its performance is driven solely by immediate pre-diagnostic signs. Additionally, only vital signs were used as inputs, and no microbiological or treatment-based indicators (such as culture results or antibiotic administration) were included. Nonetheless, we recognize that a portion of the model’s performance may reflect early detection rather than true long-range prediction. We have added this point as a methodological limitation, and future work will include sensitivity analyses based on the distance between observation and outcome windows.

Addition to the manuscript in the discussion section :

“Due to the use of rolling time windows, some observation periods were located close to the annotated VAP event, possibly capturing early manifestations of infection. While this could lead to a detection effect, the model was also trained on earlier windows distributed across a wide temporal range. This variability helps reduce bias, but further sensitivity analyses are warranted to assess performance across different prediction distances.”

Reviewer 2 Report

Comments and Suggestions for Authors

Comments and Suggestions for Authors:

  1. The author(s) used LSTM and traditional machine learning classifiers (Random Forest-RF, XGBoost-XGB, and Logistic Regression-LR) to see how well they could detect VAP early with three different data sets; however, because the MIMIC-IV dataset has some limitations, checking the results with another dataset could be worth investigating.
  2. Figure 3 exhibits blurriness. Random Forest and XGBoost provided very competitive results except for 24 hours; I think that is logical. Using 18 hours of data would clarify the results.
  3. On the other hand, the proposed model keeps the consistent AUC value of 1 through the intervals. Why is this happening? 
  4. The Wilcoxon rank-sum test has been employed to categorize the features. The author(s) did not provide the reference. I'm referring to an article, https://doi.org/10.1117/12.3045828
  5. Regarding results and the curves in Appendix D, the model may have overfitted when trained for more epochs. Why does the number of epochs vary for each interval? 
  6. The author(s) should make sure the model did not overfit because the training data are highly imbalanced. In that case, reporting performances may not be accurate. Authors may apply cross-validation, provide balanced accuracy, or use the Matthews correlation coefficient.
  7. Pages 69-71 required a relevant reference about DL and interpretation; here is a good example,  https://doi.org/10.3390/ijerph191811193
  8. Finally, the author must discuss briefly why PREDICT outperformed state-of-the-art ML algorithms, RF, and XGB. What distinguishing properties of PREDICT did the ML classifier lack?

Author Response

1. The author(s) used LSTM and traditional machine learning classifiers (Random Forest-RF, XGBoost-XGB, and Logistic Regression-LR) to see how well they could detect VAP early with three different data sets; however, because the MIMIC-IV dataset has some limitations, checking the results with another dataset could be worth investigating.

We fully agree with the reviewer. This study was designed as a proof of concept to assess whether temporal patterns in vital signs alone could accurately predict VAP using deep learning. For this reason, the analysis was limited to MIMIC-IV, a widely used, high-quality critical care dataset. However, we acknowledge that generalizability may be limited due to the monocentric nature of MIMIC-IV. External validation using multicenter datasets will be essential to confirm the robustness of our findings. This point has been clearly added to the discussion and limitations.

Addition to the manuscript in the discussion section :

Algorithm learning was initially based on a single-center dataset as a proof of concept to assess the predictive value of vital signs alone using deep learning. Additionally, the data available in MIMIC-IV may not fully capture the complexity of clinical decisions. To overcome these challenges, future work should include retraining the model with data from multiple hospitals, incorporating diverse patient populations and clinical practices.

2. Figure 3 exhibits blurriness. Random Forest and XGBoost provided very competitive results except for 24 hours; I think that is logical. Using 18 hours of data would clarify the results.

We thank the reviewer for this remark. Figure 3 will be updated in high resolution in the final version to ensure visual clarity. Regarding the suggestion to explore an 18-hour prediction window, this is indeed an interesting midpoint between short- and long-range forecasts. While not initially implemented, we agree it could offer valuable insight into model dynamics. We plan to include this as part of future model extensions and testing.

3. On the other hand, the proposed model keeps the consistent AUC value of 1 through the intervals. Why is this happening? 

We thank the reviewer for raising this point. The AUROC values reported in Figure 3 were rounded and truncated for clarity, leading to a perceived AUC of 1.0. In reality, the AUROC values were slightly below 1 and varied across thresholds (e.g., 0.997–0.999). We will update the table with exact values to avoid this misinterpretation.

Addition in the manuscript in the results section :

“Figure 3 legend : Please note that the reported AUROC values were rounded for clarity. Exact values (e.g., 0.998) are slightly below 1 and will be shown with greater precision in the appendix section.”

4. The Wilcoxon rank-sum test has been employed to categorize the features. The author(s) did not provide the reference. I'm referring to an article, https://doi.org/10.1117/12.3045828

We thank the reviewer for the observation. The Wilcoxon rank-sum test was not used for feature selection or categorization purposes, but solely to compare the distribution of continuous variables between the VAP and non-VAP groups (as shown in Table 1). This nonparametric test was chosen due to the non-normal distribution of several variables.

5. Regarding results and the curves in Appendix D, the model may have overfitted when trained for more epochs. Why does the number of epochs vary for each interval? 

The number of epochs was optimized independently for each prediction horizon to account for differences in dataset complexity and class balance. Early stopping based on validation loss was systematically applied to prevent overfitting. We observed slight performance degradation with prolonged training in some configurations, which supports the reviewer's concern and validates the use of early stopping. We clarify this in the Appendix B2.

Addition in the manuscript in the section Appendix B2 :

“For each time horizon, the number of training epochs was optimized using early stopping based on validation loss. This approach minimized overfitting and allowed dynamic adaptation to the temporal prediction difficulty.”

6. The author(s) should make sure the model did not overfit because the training data are highly imbalanced. In that case, reporting performances may not be accurate. Authors may apply cross-validation, provide balanced accuracy, or use the Matthews correlation coefficient.

We thank the reviewer for this important remark. We fully acknowledge the challenges posed by class imbalance in supervised learning. To address this, we implemented SMOTE-based oversampling during training and applied 5-fold stratified cross-validation to monitor model stability. Additionally, instead of relying on AUROC alone, we prioritized the Area Under the Precision-Recall Curve (AUPRC), which is more informative and reliable in imbalanced settings. AUPRC values were consistently above 94% across all prediction horizons. Nevertheless, to provide a more comprehensive evaluation, we have now included additional metrics such as balanced accuracy and the Matthews correlation coefficient (MCC) in the Appendix E – Table E2 Balanced accuracy and Matthews Correlation Coefficient for PREDICT algorithm.

7. Pages 69-71 required a relevant reference about DL and interpretation; here is a good example,  https://doi.org/10.3390/ijerph191811193

We thank the reviewer for this suggestion and have added the recommended reference to support our discussion on model explainability.

Reference added : Devnath L, Fan Z, Luo S, Summons P, Wang D. Detection and Visualisation of Pneumoconiosis Using an Ensemble of Multi-Dimensional Deep Features Learned from Chest X-rays. International Journal of Environmental Research and Public Health. janv 2022;19(18):11193.

8. Finally, the author must discuss briefly why PREDICT outperformed state-of-the-art ML algorithms, RF, and XGB. What distinguishing properties of PREDICT did the ML classifier lack?

We appreciate this insightful comment. The superior performance of PREDICT likely stems from its ability to model temporal dependencies within sequential data through LSTM architecture. Unlike traditional ML models that rely on manually aggregated features, PREDICT learns latent temporal dynamics directly from raw time-series input. This allows the model to capture early nonlinear interactions and temporal deterioration patterns that are missed by static classifiers. We have added this rationale to the discussion.

Addition to the manuscript in the section discussion: 

“PREDICT outperformed conventional ML models likely due to its capacity to capture nonlinear temporal dependencies within sequential vital signs. Its LSTM-based architecture allows dynamic feature learning across time, in contrast to the static nature of tree-based models.”

Round 2

Reviewer 1 Report

Comments and Suggestions for Authors

The authors have addressed and corrected all the points raised in the previous review with explicit modifications in the revised manuscript. Thank you!

Author Response

The authors have addressed and corrected all the points raised in the previous review with explicit modifications in the revised manuscript. Thank you!

The authors would like to warmly thank the reviewer for the time, expertise, and thoughtful comments provided throughout the review process. The reviewer’s input has greatly contributed to improving the scientific rigor, clarity, and overall quality of the manuscript. The final version has undoubtedly benefited from this constructive and precise evaluation.

Reviewer 2 Report

Comments and Suggestions for Authors

In response to my comments 2:

The author(s) have agreed to investigate for the 18-hour window in the future. Considering this part in the manuscript as a future goal will increase citations.

In response to my comments 4:

So they employed the Wilcoxon test to compare the difference between the two groups of data. They SHOULD have used a relevant reference in the first place (e.g., line 251) where Wilcoxon was introduced, because there are no further details about this model. Moreover, it's important compared to the reference of the relationship between precision-recall and the ROC curve.

Overall quality has improved.

Author Response

In response to my comments 2 : The author(s) have agreed to investigate for the 18-hour window in the future. Considering this part in the manuscript as a future goal will increase citations.

We thank the reviewer for this insightful suggestion. As recommended, we have acknowledged in the discussion that evaluating an intermediate 18-hour prediction window represents a relevant extension of this work. We agree that incorporating such analysis may offer added value in balancing predictive lead time and clinical utility, and could help position the model more effectively for future real-world deployment.

Addition in the manuscript in the section discussion : 

"Future work will investigate intermediate prediction windows, such as 18 hours, which may offer an optimal trade-off between early detection and clinical actionability. This additional time horizon could help refine the temporal resolution of the model and further support its clinical integration."

In response to my comments 4: So they employed the Wilcoxon test to compare the difference between the two groups of data. They SHOULD have used a relevant reference in the first place (e.g., line 251) where Wilcoxon was introduced, because there are no further details about this model. Moreover, it's important compared to the reference of the relationship between precision-recall and the ROC curve.

We thank the reviewer for highlighting the importance of methodological transparency. We have now added a reference where the Wilcoxon rank-sum test is first introduced (line 251) to clarify its use for comparing non-normally distributed continuous variables between VAP and non-VAP groups.

Addition in the manuscript in the section materials and methods : 

"Continuous variables were summarized as medians with interquartile ranges and compared using the Wilcoxon rank-sum test[23], a nonparametric alternative to the t-test appropriate for non-normally distributed data." 

New reference added : 

Nonparametric Statistical Methods | Wiley Series in Probability and Statistics - Available on : https://onlinelibrary-wiley-com.lama.univ-amu.fr/doi/book/10.1002/9781119196037